# Neuronal Stress and Injury Caused by HIV-1, cART and Drug Abuse: Converging Contributions to HAND

**DOI:** 10.3390/brainsci7030025

**Published:** 2017-02-23

**Authors:** Ana B. Sanchez, Marcus Kaul

**Affiliations:** 1Immunity and Pathogenesis Program, Infectious and Inflammatory Disease Center, Sanford Burnham Prebys Medical Discovery Institute, La Jolla, CA 92037, USA; asanchez@SBPdiscovery.org; 2Department of Psychiatry, University of California San Diego, San Diego, CA 92093, USA

**Keywords:** HIV-1, HAND, anti-retroviral, methamphetamine, neurotoxicity

## Abstract

Multiple mechanisms appear to contribute to neuronal stress and injury underlying HIV-associated neurocognitive disorders (HAND), which occur despite the successful introduction of combination antiretroviral therapy (cART). Evidence is accumulating that components of cART can itself be neurotoxic upon long-term exposure. In addition, abuse of psychostimulants, such as methamphetamine (METH), seems to compromise antiretroviral therapy and aggravate HAND. However, the combined effect of virus and recreational and therapeutic drugs on the brain is still incompletely understood. However, several lines of evidence suggest a shared critical role of oxidative stress, compromised neuronal energy homeostasis and autophagy in promotion and prevention of neuronal dysfunction associated with HIV-1 infection, cART and psychostimulant use. In this review, we present a synopsis of recent work related to neuronal stress and injury induced by HIV infection, antiretrovirals (ARVs) and the highly addictive psychostimulant METH.

## 1. Introduction

Thirty-five years ago infection with human immunodeficiency virus (HIV-1) and acquired immunodeficiency syndrome (AIDS) evolved into an acute epidemic [1,2]. However, the development and use of combination antiretroviral therapy (cART) since the mid-1990s has changed the course of HIV infection/AIDS, which is now considered a chronic and treatable disease, if therapy is accessible [3]. Still, each year, more than 1 million people die from AIDS-related causes and 2.1 million people become newly infected by HIV-1. Globally, an estimated 36.7 million persons were living with HIV/AIDS in 2015, including more than 1.2 million people in the USA alone [4,5,6]. HIV-infection of the central nervous system (CNS) often leads to neurological problems and HIV-associated neurocognitive disorders (HAND) [7]. The underlying neuropathological mechanisms in humans are not completely understood and persist despite the use of antiretrovirals (ARVs) and effective virological control [7,8,9,10,11]. A major comorbidity of HIV infection is the abuse of drugs, such as heroin, cocaine and methamphetamine (METH), which is a public health problem in its own right [12]. Therefore, opiates, cocaine and METH are all being intensely studied in order to better understand their effects on HIV infection and HAND [13,14,15,16,17,18,19], but for the purpose of this review we will focus on the role of METH as one example of a widely abused drug. As such, HIV infection is frequently linked with the recreational use of the psychostimulant METH [20,21,22] and diminished adherence to cART regimens [23]. Additionally, increased viral loads have been linked to METH use in ART–receiving HIV positive individuals. [23,24]. The combination of METH and HIV-1 appears to trigger more neurocognitive impairment and neuropathology than either agent alone but the presumed mechanistic interaction of virus, anti-viral treatment and the psychostimulant drug is incompletely understood [25,26,27,28].

Clearly, the incidence of HIV-associated dementia (HAD), the most severe form of HAND, has declined with the implementation of cART [29,30], but the prevalence of cognitive impairment milder than dementia remains high in HIV patients on cART [9,11,31]. In addition, recent clinical studies observed that discontinuation of cART in virologically controlled HIV patients unexpectedly resulted in significant improvement of neurocognitive function [32,33,34,35]. Furthermore, experimental evidence is accumulating that at least some ARV compounds may themselves have neurotoxic effects [36,37,38,39,40]. These observations raised the possibility that certain ARVs may have neurotoxic effects that could contribute to the development of HAND.

Therefore, METH-using HIV patients are at risk of exposure to a combination of potential contributors to neurotoxicity: HIV and its components, a psychostimulant drug and ARVs. Here we will discuss current information about those contributing factors, the combination of which is being encountered in the clinical setting while an understanding of the mechanisms at the cellular level is only beginning to emerge.

## 2. HIV Infection, Neurotoxicity and HAND

Humans of all ages infected with HIV-1 develop neurological symptoms that include motor and cognitive dysfunction which are now termed HIV-associated neurocognitive disorders (HAND) [7,41,42,43]. HAND defines three categories of disorders with increasing severity according to standardized measures of dysfunction: (i) asymptomatic neurocognitive impairment (ANI); (ii) mild neurocognitive disorder (MND) and (iii) HAD. With the advent of cART, the incidence of HAND’s most severe form, dementia, decreased, suggesting a beneficial effect on cognitive function [29,30,44]. However, the prevalence of milder cognitive impairment (ANI, MND) remains high in HIV patients on cART, and HAND/HAD continues to be a significant independent risk factor for death due to AIDS [9,11,31,45]. Although improved control of peripheral viral replication and the treatment of opportunistic infections continue to extend survival times, cART fails to provide protection from HAND, or to reverse the disease in most cases [46,47,48,49,50,51]. One study found that in a group of 669 HIV patients who passed away between 1996 and 2001 more than 90% had developed HAD as an AIDS-defining condition during the last 12 months of life [52]. Moreover, the proportion of new cases of HAND/HAD displaying a cluster of differentiation 4^+^ (CD4^+^) T cell count greater than 200 μL^−1^ is growing [50]. This situation and distinct patterns of viral drug resistance in plasma and cerebrospinal fluid (CSF) compartments might at least in part be explained by limited penetration into the CNS of HIV protease inhibitors and several of the nucleoside analogues [46,53,54]. Therefore, it is possible that as people live longer with HIV-1 infection the prevalence of dementia may continue to rise [43,46,50,53,54,55,56].

Pathological features of HIV-1 infection in the brain are often referred to as HIV encephalitis (HIVE) and include activated resident microglia, microglial nodules, multinucleated giant cells, infiltration predominantly by monocytoid cells, including blood-derived macrophages, and decreased synaptic and dendritic density, combined with selective neuronal loss, widespread reactive astrocytosis, and myelin pallor [57,58]. However, the pathologic features best correlating with *ante mortem* measures of cognitive dysfunction include increased numbers of microglia [59], decreased synaptic and dendritic density, selective neuronal loss [58,60,61], elevated tumor necrosis factor (TNF)-α mRNA in microglia and astrocytes [62], and evidence of excitatory neurotoxins in CSF and serum [63]. Furthermore, two reports provide evidence that the amount of proviral HIV DNA in circulating monocytes and macrophages correlates better than viral load with the risk of developing HAD [64,65].

HIV infection can be associated with neuronal damage and loss in distinct brain regions, including frontal cortex [66,67], substantia nigra [68], cerebellum [69], and putamen [70] and features of neuronal apoptosis have been found in brains of HAD patients [71,72,73]. Moreover, the localization of apoptotic neurons was correlated with signs of structural damage and closely associated with evidence of microglial activation, especially within subcortical deep gray structures [71].

With the introduction of cART, HIV neuropathology began to shift. Although the incidence of opportunistic infections seemed to decline, two *post mortem* studies observed increased macrophage/microglia infiltration and activation in hippocampus and basal ganglia of cART-treated HIV patients as compared to samples from the pre-cART era as well as a higher prevalence of HIVE at the time of autopsy [25,74]. Specimens from HIV patients who had failed cART displayed even more encephalitis and severe leukoencephalopathy [74]. In line with these reports are more recent neuropathological descriptions of various forms with severe HIVE and white matter injury, extensive perivascular lymphocytic infiltration, “burnt-out” forms of HIVE and seemingly aging-related beta-amyloid accumulation implying an Alzheimer’s-like neuropathology [75,76].

HIV-1 appears to reach the brain soon after infection in the periphery, and then localizes primarily to perivascular macrophages and microglia [77,78,79,80]. Infection by HIV-1 of macrophages and lymphocytes in the periphery and microglia in the brain occurrs after the viral envelope protein gp120 binds to CD4 in conjunction with at least one of several possible chemokine receptors. Depending on the viral strain, different HIV-1 variants use CC chemokine receptor 5 (CCR5, CD195) and CCR3, or CXC chemokine receptor 4 (CXCR4, CD184), or a combination of these chemokine receptors to enter target cells [81,82,83]. Neurons and astrocytes in the brain also express chemokine receptors, including CCR5 and CXCR4 [84,85]. However these cells, in contrast to microglia, appear to be largely refractory to productive HIV-1 infection under in vivo conditions. However, several in vitro studies strongly suggest that CXCR4 is prominently involved in HIV-associated neuronal damage whereas CCR5 may play a dual role by being able to either serve a toxic or protective function [86,87,88,89,90,91,92]. Intact HIV-1, as well as picomolar concentrations of isolated viral envelope gp120, can induce neuronal death via CXCR4 and CCR5 receptors in neurons from humans and rodents [86,87,89,90,93,94,95,96,97,98].

While progress is being made in characterizing the neuropathological processes, how exactly HIV-1 infection provokes neuronal injury and death as well as neurocognitive and motor deficits remains controversial [43,53,54,79,99]. While it is generally agreed upon that HIV-1 does not infect post-mitotic, mature neurons, the mechanism of neuronal damage is a matter of debate and continuing investigation. Ample evidence exists that various viral proteins; including Tat, Nef, Vpr and the Env proteins gp120 and gp41, can initiate neuronal injury and death [43,53,93,99,100,101,102,103,104]. Moreover, we and others found more recently that HIV-1 and at least its gp120 can also compromise neurogenesis [8,105,106]. All these observations, in particular those related to neurotoxicity, have contributed to at least two different possible explanations of how HIV-1 initiates brain injury, the “direct injury” and the “indirect” or “bystander effect” hypothesis. These two hypothetical mechanisms are by no means mutually exclusive, and the available data suggest a role for both. However, under conditions where glial and neuronal cells are present, the indirect neurotoxicity mediated by macrophages and microglia may predominate [43,53,78,79,92,99,107,108,109].

The hypothesis that HIV proteins can directly injure neurons without any contribution of non-neuronal cells (microglia/macrophages and/or astrocytes) is supported by experiments showing that viral envelope protein gp120, Tat, and Vpr are toxic in serum free primary neuronal cultures [87,88] or in neuroblastoma cell lines [86,99,102]. The absence of non-neuronal cells permits the investigation of potential direct effects of viral proteins on neurons, but a predominantly indirect effect cannot be detected.

In mixed neuronal/glial cultures that recapitulate the cellular composition of the brain, HIV-1, gp120 and Tat appear to induce neuronal apoptotic death primarily in an indirect fashion via the release of toxins from macrophages/microglia [89,93,94,95,97,98,107,109,110]. Moreover, both the inactivation or depletion of macrophages and microglia basically abrogate neurotoxicity of HIV-1/gp120 in mixed neuronal/glial cultures [89,98,107,111] and it appears likely that stimulation of CXCR4 or CCR5 in macrophages/microglia is a prerequisite for the neurotoxicity of HIV-1 and gp120 [89,96,109].

At least some of the neurotoxins released by HIV-1 infected or gp120-stimulated macrophages/microglia stimulate an ionotropic glutamate and neurotransmitter receptor, the *N*-methyl-d-aspartate-type receptor (NMDAR) [97,112,113]. NMDAR antagonists can prevent neuronal cell death in vitro due to HIV-infected or gp120-activated macrophages [97,112]. Activation of ionotropic glutamate receptors in neurons initiates under normal physiological conditions a transient depolarization and excitation that plays an important role in neurocognitive function [114,115]. In contrast, excessive and/or extended NMDAR stimulation causes excitotoxicity via a mechanism involving a sustained elevation of intracellular Ca^2+^ concentration, a subsequent compromise of mitochondrial function and cellular energy metabolism which in turn results in the production of free radicals [43,115,116]. In case of a mild but sustained insult, as it seems to occur in HIV infection, neurons eventually undergo programmed cell death (apoptosis) which has also been observed in *post-mortem* brains of HIVE/HAD patient [71,72,89,96]. Neuronal apoptosis triggered by HIV-1 or gp120 toxicity or a direct excitotoxic insult involves neuronal Ca^2+^ overload, activation of p38 MAPK and p53, activation of cell cycle protein, caspases, release of cytochrome c and other molecules, such as apoptosis-inducing factor (AIF) from mitochondria, free radical formation, lipid release and peroxidation, and chromatin condensation [84,89,90,98,109,117,118,119,120,121]. Alterations of cellular lipid metabolism, and an increase in ceramide, sphingomyelin and hydroxynonenal as well as the activation of the unfolded protein response (UPR) have also been linked to oxidative processes and cellular distress in the neurotoxic pathways associated with HIV infection [99,122,123].

## 3. Neurotoxicity of Antiretroviral Drugs

ARV drugs are classified based on the mechanism action and currently include (i) nucleoside/nucleotide reverse transcriptase inhibitors (NRTI); (ii) non-nucleoside reverse transcriptase inhibitors (NNRTI); (iii) protease inhibitors (PI), (iv) integrase strand transfer inhibitors (INSTI); (v) a fusion inhibitor (FI); and (vi) an entry inhibitor (EI), blocking chemokine receptor CCR5. In addition, two drugs, ritonavir (RTV) and cobicistat (COBI) are used as pharmacokinetic enhancers [124] (reviewed in [125,126]). The Department of Health and Human Services (HHS) Panel on Antiretroviral Guidelines for Adults and Adolescents (a working group of the office of AIDS research) [124] keep updating the combination regimens, based on more than 25 antiretroviral drugs approved by the Food and Drug Administration (FDA) in order to guide to the right and best treatment for HIV^+^ individuals. Currently recommended cART regimen consist of two NRTIs (usually abacavir/lamivudine (ABC/3TC), tenofovir alafenamide/emtricitabine (TAF/FTC) or tenofovir disoproxil fumarate/emtricitabine (TDF/FTC)) in combination with a third active ARV drug from one of three drug classes: INSTI, NNRTI or PI with a pharmacokinetic enhancer (RTV or COBI). Other combination and/or regimen are selected based primarily on antiviral efficacy, potential adverse effects (toxicity), pill burden, drug-drug interaction, comorbid conditions and cost (see [124] on 12/1/2016). As a result of effective cART, HIV-1 infection has become a chronic condition and infected individuals have a longer, near normal life span leading to aging with HIV as a new clinical phenomenon [51,127,128,129].

The introduction of cART clearly led to a decline in the incidence of the most severe form of HAND, HAD [29,30]. However, the occurrence of cognitive impairment milder than HAD persists in HIV patients on cART [9,11,31] and recent studies observed that discontinuation of cART in virologically controlled HIV patients unexpectedly resulted in significant improvement of neurocognitive function [32,33,34,35]. Furthermore, clinical and experimental evidence is accumulating that at least some ARV compounds may themselves have neurotoxic effects [36,37,38,39,40,125,130].

Certain ARV or combinations thereof (cART) have been reported to be toxic in the periphery but also the brain, which could contribute to the development of HAND [32,36,37,38,39,40,131,132]. Several other studies have provided direct evidence that anti-retroviral drugs can exert neurotoxic effects in connection with oxidative stress, dysregulation of Ca^2+^ homeostasis and alteration of mitochondrial respiration [39,40,133,134,135,136,137,138,139].

NRTIs: These compounds have primarily been linked to peripheral neuropathy, in particular didanosine (ddI), stavudine (d4T) and zalcitabine (ddC) [140]. The main unintentional target appears to be mitochondrial polymerase γ, an enzyme required to maintain mitochondrial DNA (mtDNA) in axons and Schwann cells [141,142]. Examples of NRTIs that seem less or non-toxic in terms of peripheral neuropathy or cellular neurotoxicity are emtricitabine (FTC), lamivudine (3TC), tenofovir (TAF or TDF) and abacavir (ABC) [39]. The earliest anti-retroviral drug zidovudine (AZT) has also been linked to mitochondrial toxicity, impaired neurogenesis and damage to neuronal dendrites and presynaptic terminals [131,143,144,145]. Overall, CNS neurotoxicity of NRTIs seems limited but also compound- and cell-specific [39,125].

NNRTIs: Compounds, such as rilpivirine (TMC278) and delavirdine (DLV) are considered to be non-toxic in the CNS [125]. Etravirine (ETR) recently appeared to be clinically safe but was found to be neurotoxic in vitro [39,146]. However, Efavirenz (EFV) and nevirapine (NVP), which are FDA approved and used in NNRTI regimen [124], have been found to exert neurotoxicity [39,137,147]. EFV has also been linked in the clinical setting to deterioration of neurocognitive function [127,130,148]. One study found that EFV caused in neuron-like SHSY-5Y cells and primary rat striatal neurons a loss of ATP, depolarization and fragmentation of mitochondria and increased mitophagy and autophagy in general suggesting disturbance of energy homeostasis as a mechanism of toxicity [149,150]. EFV was also found to cause endoplasmic reticulum (ER) stress in human brain endothelial cells and in microvessels of the CNS in HIV-transgenic mice [151]. However, the latter study found that EFV inhibited autophagy by binding to a complex comprising Beclin 1, autophagy-related 14 (ATG14) and Phosphatidyl inositol 3 kinase III (PI3KIII) which is required for formation of an autophagosome. In addition, an enzyme of the cytochrome P450 (CYP) family, CYP2B6, affects the metabolism and therefore likely the concentration of EFV in the brain, which may contribute to functional impairment in the CNS as the 8-hydroxy metabolite of EFV has been found to be neurotoxic [148,152].

PIs: This group of ARV is very effective and part of most first line treatment regimen [153]. Saquinavir (SQV) and nelfinavir (NFV) were also found to be very well tolerated in HIV patients whereas ritonavir (RTV) was associated with adverse and toxic effects [154]. PI, NRTI, INSTI and the CCR5 blocker maraviroc are metabolized by enzymes of the CYP450 family. In fact, CYP450 enzymes are known to be involved in the metabolism/activation/inactivation of the majority of pharmaceutical drugs [155]. Most of the human CYP450s are primarily membrane-associated proteins located in the inner membrane of the mitochondria or in the endoplasmic reticulum of cells. Especially the subclasses CYP2 and CYP3 are involved in the metabolism of drugs and steroids, which explains the risk of drug-drug interactions in cases of polypharmacy, such as cART. However, those interactions can on occasion be harnessed for therapeutic purposes. As such RTV has been used as a booster for mono- or triple therapy [156]. Other PIs, namely amprenavir (APV), indinavir (IDV) and atazanavir (ATV) have also been implicated in neurotoxicity [157,158,159]. Moreover, a recent study demonstrated in non-human primates, rodents and in vitro neuroglial cell cultures that the PIs SQY and ATV in combination with the NRTI tenofovir (TDF) and an INSTI and RTV and SQV separately caused endoplasmic reticulum (ER) stress, activated β-site amyloid precursor protein cleaving enzyme-1 (BACE-1) and neuronal damage [160].

INSTIs: Several reports have indicated possible CNS toxicity for raltegravir (RAL) and elvitegravir (EVG) as the compounds triggered clinical psychiatric symptoms [161,162,163].

FI: The only FDA approved compound, enfuvirtide (T-20) has not displayed any conclusive evidence for neurotoxicity. Some earlier studies suggested sensory neuropathy [164,165] while others found no signs of toxicity [164,166,167].

EI: The CCR5 blocker maraviroc (MVC) is the only FDA-approved compounds in its class. There is currently no evidence of neurotoxicity, and on the contrary, several studies found evidence for a neuroprotective effect of MVC [39,92,168,169,170].

As discussed above, certain ARV and cART carry the risk of neurotoxicity. One of the major concerns using cART is the role of drug-drug interaction via various CYP450 enzymes, which could aggravate neurotoxicity and contribute to the development of HAND. Therefore the need exists for a better understanding of the mechanism of not only individual effects of ARV but their combinations in order to identify the most effective and least toxic cART for HIV-infected individuals.

## 4. Neurotoxicity of METH

METH is an addictive psychostimulant drug, and its abuse can cause a number of acute and chronic symptoms, including agitation, anxiety, paranoia, psychosis and aggression [20,22,171], a variety of cardiovascular problems [172,173], reactive microgliosis [174], and hyperthermia and convulsions [175]. Abuse of METH is also associated with behavioral symptoms, including increased engagement in high-risk activities, such as unprotected sex [20,21,22,176,177]. Neurocognitive sequelae of chronic METH abuse include deficits in attention, working memory and executive functions [28,176,178,179,180,181]. In combination, METH and HIV-1 appear to cause more neurocognitive deficits than either agent alone, but the potential mechanistic interaction of both the virus and the drug is still poorly understood [14,26,27].

The neuropathology resulting from METH abuse in humans includes a decrease of dopamine (DA) transporter (DAT) in cortex, caudate-putamen and other brain regions [182], reduction of DA and D2DA receptor in caudate-putamen [183,184], decreased density of serotonin transporter (5-HTT) in cortex [185], and abnormal glucose metabolism [186]. A pronounced interference with the nigrostriatal dopaminergic neuronal system suggests a far reaching effect on dopamine-rich fronto-striatal-thalamo-cortical circuits [187]. Moreover, a loss of gray matter in cingulate, limbic and paralimbic cortices, a significantly smaller hippocampus, and a hypertrophy of white matter and pronounced microglial activation occur as a consequence of METH use and point to pathological effects beyond the dopaminergic system [188,189].

In agreement with the findings in humans, animal studies show that METH injures presynaptic DA and 5-HT terminals [190,191], causes a decrease of tyrosine hydroxylase (TH) and tryptophan hydroxylase (TPH) [192], depletes DA and 5-HT [193], reduces DAT, 5-HTT [194,195,196] and vesicular monoamine transporter (VMAT)-2 [197]. METH also triggers neuronal cell death in cortex [198,199], striatum [198], hippocampus [200] and olfactory bulb [201]. In the striatum, METH destroys specifically gamma-amino butyric acid (GABA)-ergic neurons that express enkephalin, but not those containing substance P, neuronal nitric oxide synthase (nNOS) or cholinergic markers [202,203].

The psychostimulant effect of METH is believed to result from an elevated extracellular DA concentration in the striatum due to increased release and reduced reuptake into vesicles [179,204,205,206]. Possibly via the increase of DA, METH appears to indirectly affect glutamatergic, GABAergic and serotonergic neurotransmission. One possible link involves striatal medium spiny neurons which receive synaptic input from nigrostriatal DA-ergic and corticostriatal glutamatergic neurons [179,207].

On the other hand, in macrophages and dendritic cells, METH compromises phagocytosis and antigen processing, important processes in building and maintaining a functional and protective immune response [28,178,179,208,209,210].

Other cellular effects of METH in the brain are reflected by signs of oxidative stress, neuroinflammation and apoptosis [211,212]. Abuse of METH seems to cause excessive neuronal release of monoamine neurotransmitters, particularly DA, in the synapse, which can be toxic to nerve terminals [213,214]. As a result of DA accumulation, levels of free radicals increase inside neurons, dysfunction of the ubiquitin-proteasome system ensues, oxidative protein nitration occurs, endoplasmic reticulum stress (ERS) and expression of p53 and inflammatory cytokines is induced, and microtubules may be deacetylated [215,216,217,218]. All these processes promote neurotoxicity via damage and/or dysfunction of proteins or cellular organelles, resulting in up-regulation of autophagy, a vital homeostatic mechanism required for maintenance of healthy and functional neurons [215,216,217,218].

## 5. Neuronal Injury by HIV + cART + METH

HIV infection is frequently linked with the recreational use of drugs, such as the psychostimulant METH [20,21,22,219]. Moreover, the use of METH increases the risk of infection with HIV-1 [20,21,22]. Recent clinic-based surveys found that depending on the demographics between 16% of heterosexual men and 15% of women and up to 20% to 50% of HIV-infected men who have sex with men (MSM) reported METH use over the past 12 months [21,219,220]. Thus, the brain of many HIV patients is exposed to HIV-1, combinations of ARVs and psychostimulants, such as METH, at the same time.

Elevated viral loads have been linked to METH use in ART–receiving HIV positive individuals [23,24] and METH-users with HIV also have shown greater neuronal injury compared with patients who do not abuse the drug [25,27]. The combination of METH and HIV-1 seems to cause more neurocognitive deficits and neuropathology than either agent alone [28,219,221]. We and others have been able to recapitulate some of those combined effects in animal models [219,222,223,224,225]. However, the potential mechanistic interaction of virus and psychostimulant drug is still incompletely understood and further complicated by the multitude of cART regimen [28,145,219,221].

Therefore, our group set out to investigate in vitro the potential contribution of all these factors to neuronal injury and loss [145]. In our study, we exposed mixed neuronal-glial cerebrocortical cells to ARVs of four different pharmacological categories (NRTI: AZT, NNRTI: NVP, PI: SQV and INSTI 118-D-24) with and without METH, and in some experiments HIV-1 gp120, an established neurotoxicity inducing viral protein. The incubations lasted for 24 h and 7 days in order to assess more acute, short-term and long-term effects, respectively. Subsequently, we assessed neuronal injury using different approaches. First, the exposed cell cultures were analyzed by fluorescence microscopy using specific markers for neuronal dendrites and pre-synaptic terminals; Second, we analyzed the disturbance of neuronal ATP levels; and third, we assessed involvement of autophagy using immunofluorescence and Western blotting. We found that ARVs led to alterations of neurites and presynaptic terminals predominantly during the 7-day incubation and depending on the specific compounds and their combinations with and without METH. Similarly, specific drug combinations with and without METH or viral gp120 as well as METH and gp120 each alone, all caused a significant loss of neuronal ATP, but none of the ARV applied as a single drug. Loss of ATP was accompanied by activation of adenosine monophosphate-activated protein kinase (AMPK) and autophagy, which, however, did not suffice to restore neuronal ATP to normal levels. In contrast, enhancing autophagy with rapamycin averted the long-term drop of ATP during exposure to cART in combination with METH or gp120.

Our study yielded a number of unexpected findings pointing to the complexities of the overall effects of exposure to HIV + cART + METH [145]. The psychostimulant drug, cART and HIVgp120 each alone and in combination led to a similar loss of neuronal ATP but lacked any additive effect. Moreover, METH or HIV/gp120 or an ARV each could exert neurotoxicity in terms of compromising neuronal dendrites and presynaptic terminals or diminishing neuronal ATP or, in the case of gp120, triggering the loss of neurons themselves when applied separately, but not necessarily when combined with ARV. Certain ARV combinations did not affect neuronal dendrites or presynaptic terminal in any detectable fashion in the presence or absence of METH, yet caused significant reduction of neuronal ATP levels. Thus, neurons were able to maintain their overall dendritic and presynaptic structures despite a disruption of neuronal energy homeostasis, an event that may explain, at least in part, progressive neurological diseases, such as HAND. Strikingly, METH, which itself is toxic to neurons [28], lacked any detectable damaging effect on neurites and synapses in combination with several ARVs and even ameliorated a significant reduction of neuronal ATP in combination with a cocktail of four ARVs. However, the admixture of four therapeutic compounds and psychostimulant compromised neuronal dendrites and synapses during long-term exposure, an effect that has also been reported by others as a result of ARV treatment [39]. Moreover, diminished neuronal ATP levels resulted when four ARVs and METH were combined with HIV-1 gp120.

All those observations were in line with earlier reports that ARVs can compromise mitochondria at various levels by altering (i) mitochondrial membrane components, such as transporters; (ii) mitochondrial kinases, such as Thymidine kinase 2 (TK2) and deoxyguanosine kinase (dGK) [138]; (iii) mitochondrial bioenergetics, in particular membrane potential; and (iv) mitochondrial DNA homeostasis by inhibiting polymerase γ [134,135,136] or promote major deletions in neuronal mtDNA [133,139]. Interestingly, our findings also suggested that neuronal energy homeostasis could significantly change without causing overt structural neuronal damage or loss. The regulatory mechanism allowing for such neuronal adaptation remains to be elucidated.

Mitochondria are the cellular organelles supplying energy in the form of ATP. The lack of ATP as an energy source could be one of the factors that compromise neuronal function before cell death occurs. Neurons are known to produce energy almost exclusively through mitochondrial oxidative phosphorylation (OXPHOS). In contrast, astrocytes can stimulate glycolysis by activation of 6-phosphofructo-2 kinase (PFK2) through AMPK in order to maintain ATP levels. However, expression of PFK2 is low in neurons rendering this pathway insufficient [226,227]. Neuronal stress, such as the above discussed excitotoxicity or disturbance of the ER, may cause a disruption of mitochondrial homeostasis and consequently diminish OXPHOS. This situation is often associated with production of more reactive oxygen species (ROS) than the intracellular redox system can detoxify and precedes neurite retraction and eventual cell death [228]. Oxidatively damaged mitochondria undergo constant remodeling and turnover in neurons, a process that also demands energy [229], as functional mitochondria are transported along the microtubules to remote axonal regions [230], and conversely dysfunctional mitochondria are returned to the cell body for degradation [231]. The induction of endogenous anti-oxidant factors can apparently block the toxicity of ARV drugs, but it is not known if those protective measures are associated with normal neuronal ATP levels [40]. Maintenance of a functional population of mitochondria is critical for postmitotic neurons, and is regulated mainly by autophagy of the damaged organelles, a process called mitophagy [232,233]. This clearance usually contributes to maintenance or the re-establishment of baseline cellular ATP levels [234,235] and is mainly controlled by AMPK and mTOR signaling pathways [236]. Since ARV drugs, METH and HIV-1 or its fragments, including gp120, all generate in neurons oxidative stress and loss of ATP, activation of autophagy seems to provide a suitable protective mechanism [40,145,216]. The importance of the cellular clearance process is supported by several studies showing that impaired autophagy in neurons contributes to the development of neurodegeneration [237,238].

Since neurons induced autophagy in the presence of in HIV gp120 or cART or METH and combination thereof but failed to recover homeostatic levels of ATP, we employed rapamycin to block mTOR, which is an inhibitor of autophagy [145,239]. In the presence of cART, METH and gp120, rapamycin helped to restore neuronal ATP to control levels. Thus, neurons were in principle able to restore ATP levels by autophagy but failed to do so when challenged separately with gp120 or the ARV cocktail or in the absence of rapamycin for reasons that need to be further explored. Strikingly, rapamycin also preserved normal neuronal ATP levels over 7 days in the presence of the combination of cART and gp120 without METH or the combination of METH and gp120 without cART. This outcome might be explained if cART and gp120 engaged different disturbance mechanisms to push neuronal ATP homeostasis off balance. Those mechanisms could however neutralize each other when ARVs and gp120 were combined or could be modified by the presence of METH. The loss of ATP associated with METH exposure was significantly ameliorated by rapamycin but not completely prevented [145]. A possible explanation may be related to the observation that autophagy can cause neurite degeneration in the presence of METH [216]. Thus, autophagy may not always be protective and perhaps be associated with a risk of damaging neuronal structure. Figure 1 provides a schematic summary of the various aspects of the brain’s exposure to HIV + METH + cART.

While we are beginning to untangle the combined effects of HIV + cART + METH by characterizing the associated cellular events, more work lies ahead. Studying the combination of HIV and_cART with other abused drugs, such as heroin and cocaine, seems equally important as these substances may involve other pathways and affect HIV neuropathogenesis in different ways than METH [15,16,17,18,240]. Moreover, we largely focused on neuronal injury, but METH as well as other abused drugs also affect the immune system and HIV infection itself by compromising for example the interferon response [241,242,243,244]. Opioids, cocaine and METH are highly addictive and all seem to be able to influence gene expression in neural and immune cells via mechanisms that involve short, con-coding, regulatory microRNAs and compromise control of HIV infection as well as normal neuronal function [19,245]. Moreover, accumulating evidence suggests a critical role for exosomes, globular, membranous extracellular nanovesicles in HIV infection, HAND and also drug abuse-related neurodegenerative diseases [246,247,248,249]. A detailed discussion of the additional abused substances, microRNAs and exosomes is beyond the scope of this review, but those areas of research are poised to significantly contribute to our understanding of the combined effects of HIV + cART + abused drugs on the brain and may help to develop future, improved therapies for addiction and HAND.

## 6. Conclusions

The introduction of cART has transformed HIV-1 infection into a treatable, yet chronic disease that remains associated with a high probability for development of neurocognitive impairment, now termed HAND. Accumulating evidence points to the possibility that besides HIV-1 itself long-term exposure to cART can also contribute to the occurrence of HAND. The fact that HIV-1 infection is frequently associated with the abuse or recreational psychostimulant drugs further complicates the situation. Several lines of evidence indicate that similar to HIV-1 or its components and METH, some ARV used in cART can also induce oxidative and ER stress, compromise neuronal energy homeostasis and trigger structural neuronal injury. However, even without overt structural damage, diminished ATP levels indicate that neurons may lose, at least in part, the energy reserve that is vital to maintaining their normal membrane potential and physiological function. Interestingly, neuronal energy levels can apparently drop significantly during exposure to some drugs or combinations thereof without necessarily destructing neuronal dendrites and pre-synaptic terminals, suggesting that a restoration of full neuronal function may be possible. Altogether, the available data suggest that the overall beneficial effect of cART against HIV infection can be accompanied by a discernable level of neurotoxicity that may be modified, but in the long-term primarily aggravated by the use of psychostimulant drugs, such as methamphetamine.

## Figures and Tables

**Figure 1 brainsci-07-00025-f001:**
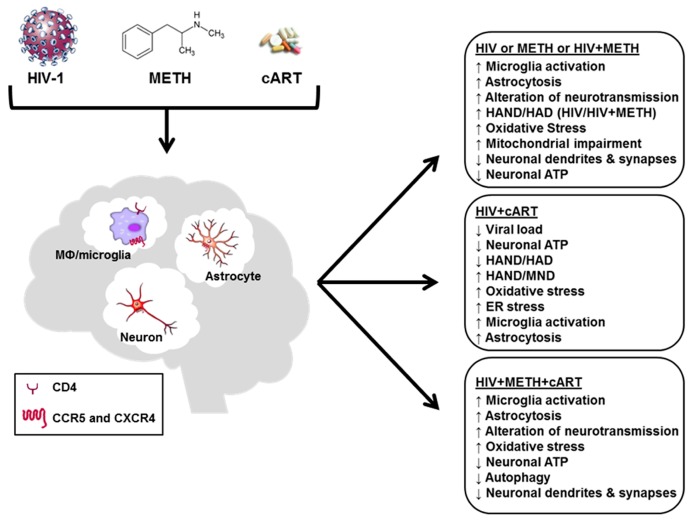
Treatment of HIV-1 infection with cART greatly reduces viral loads in periphery and the central nervous system (CNS), the incidence of HIV-associated dementia (HAD) and acquired immunodeficiency syndrome (AIDS)-related deaths. HIV-1 or its components, methamphetamine (METH) and some antiretrovirals used in combined antiretroviral therapy (cART) can induce oxidative and endoplasmic retriculum (ER) stress, compromise autophagy and neuronal energy homeostasis and trigger functional as well as structural neuronal injury. However, even without overt structural damage, diminished adenosine triphosphate (ATP) levels indicate that neurons may lose, at least in part, the energy reserve that is vital to maintaining their normal membrane potential and physiological functions, such as homeostatic regulation of neurotransmission. A compromised neuronal energy homeostasis may explain why cART permits the occurrence of HIV-associated mild neurocognitive disorders (MND). Interestingly, neuronal energy levels can apparently drop significantly during exposure to some drugs or combinations thereof without necessarily destructing neuronal dendrites and pre-synaptic terminals, suggesting that a restoration of full neuronal function may be possible.

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
