# Peer review of "Neuronal Stress and Injury Caused by HIV-1, cART and Drug Abuse: Converging Contributions to HAND"

_brainsci, 2017, doi:10.3390/brainsci7030025_

Round 1
Reviewer 1 Report
This work by Sanchez and Kaul represents great comprehension of literature highlighting how compounding effects of psychostimulant drugs like methamphetamine, antiretroviral therapy components and HIV-infection converge at inducing oxidative stress to cause neurologic injury that leads to the progression of neurocognitive disorders. This timely recognition is robust and translational in nature, which may promote important discoveries in near future. I have no major concerns.Author Response
We thank the reviewer for evaluating our manuscript and greatly appreciate the very supportive comments.
Reviewer 2 Report
This is an excellent and well written review from a respected group that documents some some significant advances in understanding how HIV exacerbates HAND in synergy with a potential psychostimulant such as methamphetamine in the era of cART. The authors describe in great detail the underlying mechanisms and what really notable is their comparison with clinical studies with the in vitro works including some from their own lab. They provide an exhaustive list of references which is really commendable in appreciating the works done earlier. I am very impressed with the overall presentation. That said, I have a a few recommendations:
If the authors can present an overall scheme in the form of a figure would be helpful to the readers.
Their sole emphasis has been meth. If the authors can correlate their analysis/findings with other stimulants especially cocaine or opiates such as morphine which have a differential effect on the neuropathogenesis would be critical and appreciated.
A small description elucidating the role other critical regulators such as miRNAs, exosomes in understanding the pathobiology would be very helpful. This can if needed be projected as future directions.
In summary, a very well written review article and I highly recommend for publication.
Author Response
We thank the reviewer for evaluating our manuscript and greatly appreciate the very supportive comments and suggestions. We agree that cocaine and opiates are equally important drugs of abuse in the context of HIV infection and revised the introduction as well as the main text to acknowledge this point (lines 36-41 and 392-406 in revised manuscript). There has been done a lot of important work on cocaine and morphine (and also alcohol and smoking/nicotine) in the context of HIV infection but a detailed discussion of that body of literature is beyond our present review. We also agree that miRNA and exosomes are important new areas of research that tremendously contribute to growing or understanding of HIV disease and consequences of drug abuse. Again, in order to not extensively expand our review we address miRNA and exosomes and their potential to further our knowledge in lines 397-406 of the revised manuscript. Also, we concur that a schematic figure may be helpful and therefore added it to the before the conclusions (line 408).
Reviewer 3 Report
This is a very thorough review written by authors who have a depth of experience in the area of METH, HIV and cognitive impairment. The review includes highlights of recent research findings from their lab on the intersection of ART, METH, and gp120 on neuronal energy homeostasis and damage.
Author Response
We thank the reviewer for evaluating our manuscript and greatly appreciate the very supportive comments.
Round 2
Reviewer 2 Report
Excellent work and all the required revisions have been duly addressed.